# Enhancing Fault Detection in Optical Networks with Conditional Denoising Diffusion Probabilistic Models

Meadhbh Healy[*1] and Thomas Martini Jørgensen[1]

[1]Technical University of Denmark
{mhea, tmqj}@dtu.dk

## Abstract

The scarcity of high-quality anomalous data often poses a challenge in establishing effective automated fault detection schemes. This study addresses the issue in the context of fault detection in optical fibers using reflectometry data, where noise can obscure the detection of certain known anomalies. We specifically investigate whether classes containing samples of low quality can be boosted with synthetically generated examples characterized by high signal-to-noise ratio (SNR). Specifically, we employ a conditional Denoising Diffusion Probabilistic Model (cDDPM) to generate synthetic data for such classes. It works by learning the characteristics of high SNRs from anomaly classes that are less frequently affected by significant noise. The boosted dataset is compared with a baseline dataset (without the augmented data) by training an anomaly classifier and measuring the performances on a hold-out dataset populated only with high quality traces for all classes. We observe a significant improved performance (Precision, Recall, and F1 Scores) for the noise affected training classes proving the success of our methods.

## 1 Introduction

Automating fault detection faces a major challenge due to the limited availability of anomalous data. Since faults are rare events in most systems, collecting a large dataset is both time-consuming and costly.

A promising solution in industrial domains is the use of synthetic data to represent fault samples for classification. Synthetic data is artificially generated but mimics real-world data. In this paper, we focus on enhancing the quantity and quality of available samples of fault classes. Optical fibers are vulnerable to various faults, both in the physical layer (e.g., fiber cuts) and from external threats (e.g., eavesdropping), which can degrade system performance. Manual fault detection requires specialized expertise and is time-intensive.

One key method for monitoring optical fibers is Optical Time-Domain Reflectometry (OTDR) [1].

OTDR works by sending pulses into the fiber and measuring Rayleigh backscattering to identify and locate faults [2]. However, OTDR trace quality can be affected by noise [3], [4], potentially leading to incorrect fault identification. The Signal-to-Noise Ratio (SNR) of the OTDR trace plays a crucial role in mitigating this issue, as low SNR traces can occur due to the fault's location or type (reflective vs. non-reflective).

We propose using a Denoising Diffusion Probabilistic Model (DDPM) to generate high-SNR OTDR traces (20-30dB) for two specific classes ("Normal" and "Bad Splice"), even though the training traces for these classes have low SNR. These two classes are chosen because of the sufficient number low-SNR samples are available in these classes for analysis. The DDPM "learns" the high-SNR characteristics by first training on four other fault classes with traces spanning an SNR range of 0 to 30dB. Afterward, the model's parameters, except for those pertaining to conditional embeddings for the signal, are frozen, and the model is retrained on the "Normal" and "Bad Splice" classes using only low-SNR traces (<5dB). Then traces from these two classes with high SNR values will be generated, having inferred high SNR traces from the original four fault classes.

To evaluate the DDPM-generated traces, we employ a machine learning (ML) classifier. The generated traces for the two classes are combined to the other four classes of real data, (between 20-30dB), and used as training data to train the ML classifier. The performance and veracity of the generated traces is measured on a holdout test set. The holdout test set consists of all six fault classes with traces with SNR values between 20-30dB. Three baselines are used for comparison: a ground truth dataset where all classes have training examples with SNR values of 20-30dB, and a sub-optimal dataset where only four classes have high-SNR samples between 20-30dB, while "Normal" and "Bad Splice" have low-SNR samples (<5dB). We compare the DDPM's performance to a Deep Convolutional Autoencoder (DCAE) trained for denoising and a conditional Variational Autoencoder (cVAE) for a generative model comparison [5]. In our case the ML classifier is used as a similarity metric to assess the veracity and fidelity of traces generated or denoised.

This approach demonstrates that the DDPM

---

*Corresponding Author.

Proceedings of the 6th Northern Lights Deep Learning Conference (NLDL), PMLR 265, 2025.

can generate high-quality OTDR traces for the specified classes, even when trained on low-SNR data, validating its effectiveness for fault detection. A workflow diagram of the process can be observed in Figure 4 in the Appendix.

## 2  Dataset Description

The dataset used in this paper is opensource [6] and consists of OTDR traces, each representing specific fault types in fiber optic network. There are six classes in total, five of which represent distinct fault classes in the optical fibre network and the sixth representing "Normal" behaviour, devoid of any of the characterized faults. The classes have approximately 16000 samples each. All classes have represented samples between 0dB and 30dB, however they are not uniformly stratified, and there can be greater or lesser amounts of low and high SNR value traces for different classes. Each observation is structured as follows:

- **Trace Sequences:** Every OTDR trace is segmented into normalized sequences, each with a fixed length of 30 data points, providing detailed insight into the fault characteristics.

- **Class:** The fault type and normal behaviour, which is one of the following six classes: "Dirty Connector", "Normal", "Bad Splice", "Reflector", "Fiber Tapping" and "PC Connector".

- **Signal-to-Noise Ratio (SNR):** The SNR value of a trace range between 0 and 30 dB - see Figure 6 in the Appendix.

- **Maximum Amplitude (Amp):** The variable 'Maximum Amplitude' denotes the maximum value observed over the trace and then divided by the position (event location). This "strength" information is for example useful for distinguishing between traces for "Dirty Connector" and "PC Connector".

The traces are inputted as a tensor of length 30 into the cDDPM, cVAE and cDCAE. The "Class", "SNR" and "Maximum Amplitude" values are embedded as vectors.

## 3  Related Work

Machine learning (ML) methods have been applied to classify OTDR traces in [7] and [8], using data with SNR levels ranging from 0 to 30 dB. While these methods perform well on the full dataset, their ability to generalize to data with SNR values below 10 dB is limited, highlighting a lack of robustness when handling unseen low-SNR data.

Generative models offer a way to create realistic and diverse data samples, closely replicating real-world scenarios, including rare fault conditions crucial for testing and refining diagnostic algorithms. Unlike other data augmentation methods, generative models not only increase data quantity but also enhance data quality, helping ML models generalize better to new, unseen samples [9].

Diffusion models, a type of generative model, have gained prominence for their ability to generate high-quality samples. In recent years, diffusion models have shown promise for generating time series data, with applications in areas such as financial forecasting and biomedical signal processing [10].

Conditioning in generative models allows the generation of data based on specific attributes, making them more flexible. This capability is particularly useful for addressing class imbalance in datasets, as it enables the generation of targeted outputs for underrepresented classes [11].

Denoising Diffusion Probabilistic Models (DDPMs) are considered state-of-the-art in generative modeling [12], though their application in AI is still emerging. For instance, Azqadan et al. used DDPMs to generate scanning electron microscope (SEM) images, producing highly realistic images and significantly streamlining the microstructure image generation process [13]. However, the use of DDPMs for generating time series data remains underexplored. Lin et al. [10] provide an overview of diffusion models for time series, discussing DDPMs, score-based generative models, and stochastic differential equations (SDEs). While DDPMs and score-based models use discrete diffusion steps, SDEs employ a continuous process, solving differential equations for data generation.

The integration of diffusion processes with other generative models is explored by Li et al. [14], where a variational autoencoder (VAE) is combined with a diffusion process to reduce aleatoric uncertainty and improve inference. This approach, applied to time series forecasting, outperforms existing models, demonstrating the power of probabilistic modeling for accurate predictions. Additionally, Adib et al. investigated synthetic time series generation for Electrocardiogram (ECG) signals using DDPMs [15]. They first converted the 1D ECG signals into 2D polar coordinates to apply computer vision techniques before feeding them into the DDPM. However, the results showed that a Wasserstein GAN [16], which processed the original 1D signals, outperformed the DDPM on all metrics. The authors suggest that future work should explore DDPMs directly on 1D signals to improve performance.

In this work, we employ a conditional Denoising Diffusion Probabilistic Model (cDDPM) to generate fault samples from rare conditions—specifically, high-SNR cases in classes that typically contain only

low-SNR faults. Rather than focusing solely on improving classification accuracy, we use the ML classifier to evaluate the authenticity and integrity of the generated traces. For comparison, we use a cDCAE, the previous state-of-the-art method for denoising OTDR traces, as proposed by Abdelli et al. [3]. Our goal is to demonstrate that generating new traces with the cDDPM, which were not part of its training set, yields better results for classification and fault detection than relying solely on denoised traces. We also use a cVAE, to compare the performance of a cDDPM for generating OTDR traces to another generative model. We aim to bridge a gap in the literature by demonstrating the potential of DDPMs not only for generating new samples, but also for producing high-quality OTDR traces that enhance fault detection.

# 4 Method

## 4.1 Preprocessing

The three conditioning embeddings, 'Class', 'SNR' and 'Maximum Amplitude' are factorized before being inputted into the embedding layer. The following datasets are created:

- **Ground Truth Dataset (GT):** This dataset contains all of the signals in each class that have an SNR value over 20dB. The counts of traces for each class is recorded in the Table 5. This is included in order to determine the ideal scenario when classifying OTDR data as it only contains samples with high SNR values.

- **Sub-Optimal Dataset (SO):** This dataset is comprised of traces from four classes; 'Dirty Connector', 'PC Connector', 'Fiber Tapping' and 'Reflector', that have an SNR value of over 20dB and two classes; 'Normal' and 'Bad Splice' that have an SNR value of under 5dB. This dataset is tested in order to emphasize the importance of SNR values classifying OTDR data. The counts of each class are recorded in Table 5 in the Appendix.

  It can be observed from Table 5 that for the classes "Fiber Tapping", "Dirty Connector", "PC Connector" and "Reflector", the number of samples in the GT dataset and SO dataset are the same. This is because for these four classes the same data is used, and only for the two analyzed classes the traces are alternated.

- **cDDPM, cVAE and cDCAE:** These three datasets is comprised of both the real traces from four classes "Fiber Tapping", "Dirty Connector", "PC Connector" and "Reflector, as well as synthetic traces generated by the cDDPM and cVAE for the "Normal" and "Bad Splice" classes. For both generative models, 1600 samples each are generated per class. For the cDCAE, the real, noisy traces are denoised and used as training samples in the ML classifier. Therefore, for the "Normal" and "Bad Splice" classes, the number of samples are 2760 and 2545 respectively.

- **Holdout Test Set:** A holdout test set is created that all the training datasets will be tested against. This contains approximately 450 samples for each class and is comprised of traces from all six classes between 20dB and 30dB.

## 4.2 ML Classifier

We design an ML classifier to distinguish between the signals for each class. The architecture of the classifier is heavily influenced by that of the BiGRU AE, originally presented by Abdelli et. al in [2]. The structure is comprised of the autoencoder consisting of GRU layers [17], followed by one fully connected layer. The GRU layers of the encoder and decoder consist of 30 and 15 neurons respectively. The fully connected layer has 16 neurons and outputs an integer between 0 and 5, depending on whatever class it classifies the fault as. The input to the classifier is a 32-length sequence; the length of the OTDR trace, the 'SNR' value, and the 'Maximum Amplitude' value of the trace. The architecture of the ML classifier can be seen in Figure 7 in the Appendix.

## 4.3 Conditioning Denoising Diffusion Model

### 4.3.1 cDDPM Process

The Conditional Denoising Diffusion Probabilistic Model (cDDPM) operates by consistently adding Gaussian noise to the data in a forward process, learning the structure of the data, and then gradually removing the noise in discrete steps to regenerate the original sample and produce new data. Training the cDDPM involves minimizing the variational upper bound on the negative log likelihood of the reverse process, aligning with a loss function that penalizes errors between the predicted and actual noise. A linear noise schedule is used for denoising, with $\beta_{min}$ set to 0.0001, $\beta_{max}$ set to 0.02, and 3000 denoising steps. The cDDPM is trained for 200 epochs. The process of training the cDDPM can be observed in Figure 5 in the Appendix.

### 4.3.2 Score Model

The noise predicted to be removed at each timestep using a neural network which we call Score Model. The architecture of Score Model involves a combination of linear and GRU layers to concentrate on the

short length of the signals. Score model consists of an input linear layer, followed by two unidirectional GRU layers and culminating in an output linear layer. The initial linear layer has a leaky ReLU [18] activation function and there is a Dropout layer between the two GRU layers to prevent overfitting [19]. The input is size 120 (the length of the sequence plus embeddings) and the first linear layer outputs 256. The first GRU layer takes 256 and increases it to 512. The second GRU layer takes an input of size 512 and decreases it to 256. The final linear layer has an output of 30. The architecture of the model can be observed in Figure 1.

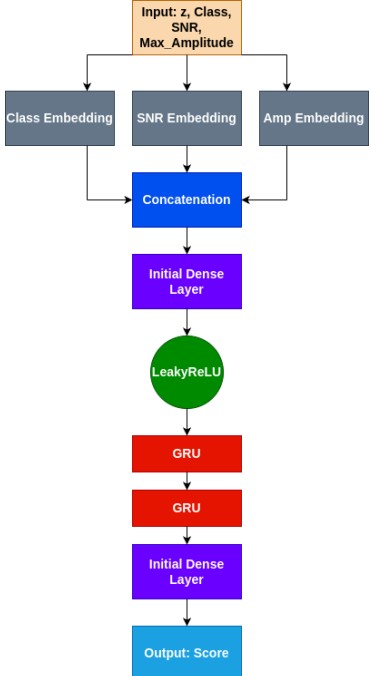

**Figure 1.** Graphic of Score Model Architecture

### 4.3.3 Conditional Embeddings

We create conditional embeddings for the parameters "Class," "SNR," and "Max Amplitude" to fully represent each OTDR trace sample under varying conditions. These embeddings provide a learnable representation for each parameter, initialized as random noise and optimized during training via backpropagation. This allows the model to learn relationships between different conditions.

The embedding sizes—3, 6, and 5 for "Class," "SNR," and "Max Amplitude," respectively—are chosen based on the relative complexity and number of categories within each condition. For instance, "Class" has 6 categories, so a size of 3 efficiently captures its variability. In contrast, "SNR," with 31 categories, requires a larger embedding size of 6 to account for its finer granularity. This proportional strategy ensures that each embedding size is

sufficient to represent the complexity of the corresponding parameter without overfitting.

### 4.3.4 Loss Function

The loss function in cDDPM targets the difference between the noise predicted to be removed by the Score Model and the actual noise used in the forward process. The Huber loss function, which balances the properties of mean squared error and mean absolute error, is used to calculate this difference. The loss function of a cDDPM is defined in Equation 1:

$$\mathcal{L} = \mathbb{E}[\mathrm{L}_\delta(\epsilon - \epsilon_\theta(x_t, t, y))] \tag{1}$$

where $x_t$: represents the data at diffusion time $t$; $\epsilon$: is the noise vector; $\epsilon_\theta(x_t, t, y)$ is the noise predicted by Score Model, conditioned by the contextual information $y$ and timestep $t$; $\mathbb{E}$ is the expectation over the distribution of the data and the forward process.

## 4.4 Conditioning Denoising Convolutional Autoencoder (cDCAE)

In order to prove the efficacy of the cDDPM for generating traces with high SNR values, we also use a cDCAE to denoise traces and compare results. The method was previously used in the work of [3] and obtained effective results denoising segmented OTDR traces. The architecture of the model can be observed in Figure 8 in the Appendix. It is similar to the architecture used by Abdelli et. al [3], with added conditional embeddings.

The cDCAE is not inherently a generative model; it is a model designed for denoising tasks. Conditioning in DCAE is used to provide additional information that can help the model better understand the context of the noise. The embeddings are inputted into both the encoder and decoder of the cDCAE. Unlike the cDDPM, it is not possible to request specific traces to be generated, instead the noisy traces are denoised and a new SNR value is computed.

## 4.5 Conditional Variational Autoencoder (cVAE)

To compare the performance of the cDDPM with the cVAE, the architecture of the cVAE is described as follows: The encoder consists of two bidirectional GRU layers with 128 and 256 neurons, respectively, followed by two fully connected layers, with a Leaky ReLU activation function between them. The fully connected layers have 256 and 128 neurons.

The decoder is composed of two unidirectional GRU layers with 256 and 128 neurons, followed by two fully connected layers, separated by a Leaky ReLU activation function, and concluding with a

Sigmoid activation function [20]. The fully connected layers in the decoder contain 128 and 64 neurons, respectively. The detailed architectures of the encoder and decoder can be seen in Figures 9 and 10 in the Appendix.

# 5 Results

All the datasets are used to train the same baseline ML classifier with all of the same tuned hyperparameters, in order to compare the quality of the signals in each dataset. We evaluate the classification of four datasets using the metrics Accuracy, Precision, Recall and F1-score.

## 5.1 Global Metrics

### 5.1.1 Accuracy

**Table 1.** Accuracy of three training datasets

| Training Set | Accuracy (%) |
|---|---|
| **Ground Truth** | 99.3007 |
| **Sub-Optimal** | 63.3304 |
| **cDDPM** | 94.4056 |
| **cDCAE** | 72.9458 |
| **cVAE** | 78.4528 |

It can be seen that the Ground Truth dataset obviously achieves the highest global accuracy with with 99.3%, as it is comprised of real traces with high SNR values. The cDDPM dataset records an accuracy 94.41%, demonstrating that the traces generated for the "Normal" and "Bad Spice" classes have a high fidelity to the real traces in GT dataset. The cDDPM dataset achieves a vastly superior performance to the cVAE demonstrating that the cDDPM is better at generating traces with high SNR values. The cDCAE dataset only marginally achieves more accuracy than the Sub-Optimal dataset, comprised of noisy traces, with 72.95% and 63.33% respectively. This illustrates that the cDCAE has failed to denoise the traces sufficiently in order to classify the test set.

## 5.2 Per-Class Metrics for GT Dataset

The metrics for all classes for the Ground Truth dataset are recorded in table 2.

| Class Label | Precision | Recall | F1 |
|---|---|---|---|
| **Normal** | 0.9966 | 0.9966 | 0.9966 |
| **Fiber Tapping** | 0.9849 | 0.9975 | 0.9912 |
| **Bad Splice** | 0.9905 | 0.9858 | 0.9882 |
| **Dirty Connector** | 0.9869 | 1.0000 | 0.9934 |
| **PC Connector** | 1.0000 | 0.9806 | 0.9902 |
| **Reflector** | 1.0000 | 1.0000 | 1.0000 |

**Table 2.** Performance Metrics by Class Label

We can observe the high precision, recall and F1 scores for all classes in the Ground Truth dataset achieve a good performance. This illustrates that none of the classes are inherently difficult or problematic to classify, provided the optimal samples are available.

## 5.3 Comparison of Datasets

The performance of each dataset is measured using precision, recall and F1 score to determine how well each class in the training set can be matched to the real traces in the test set. We also provide the Precision Recall Curve for both synthesized classes, to acquire a threshold independent estimate of the models ability to identify the real traces correctly.

### 5.3.1 Normal

The performance metrics for Ground Truth dataset and Sub Optimal dataset as well as for the cDDPM, cVAE and cDCAE are recorded in Table 3.

**Table 3.** Performance metrics for Normal

| Class | Precision | Recall | F1 Score |
|---|---|---|---|
| **Ground Truth** | 0.9966 | 0.9966 | 0.9966 |
| **Sub-Optimal** | 1.0000 | 0.0000 | 0.0000 |
| **cDDPM** | 1.0000 | 0.9024 | 0.9487 |
| **cDCAE** | 0.6750 | 0.2727 | 0.3885 |
| **cVAE** | 0.9801 | 0.9933 | 0.9866 |

It can be observed that the Ground Truth achieves the highest scores for this class, however, this is closely followed by the cDDPM. Ground Truth and Sub-Optimal as well as the cDDPM all achieve a precision of 1.0000, meaning that this class is always correctly predicted in each of these datasets. In the case of Sub-Optimal this result is achieved because the class is never predicted incorrectly. The recall and F1 scores of Ground Truth, cDDPM and cVAE are both extremely high achieving over 90% in all three. The cVAE achieves higher recall than the cDDPM for this class meaning that the ML classifier is misclassifying fewer instances of generated cVAE traces. The Sub-Optimal dataset of noisy traces fails to predict any traces in this class correctly. The cDCAE achieves inadequate results with an F1 Score of 0.3885, and is vastly outperformed by the cDDPM and cVAE, recording an F1 Score 0.9487 and 0.9866 respectively. The performance of each dataset for classifying the "Normal" class against the other classes is plotted in the PR curve in 2. The PR curve for the "Normal" class indicates that the cVAE model performs very well, nearly matching the performance of Ground Truth dataset. The cDDPM model has a slightly steeper drop-off compared to cVAE and GT, indicating that it classifies more false positives than GT and cVAE. GT, cVAE and

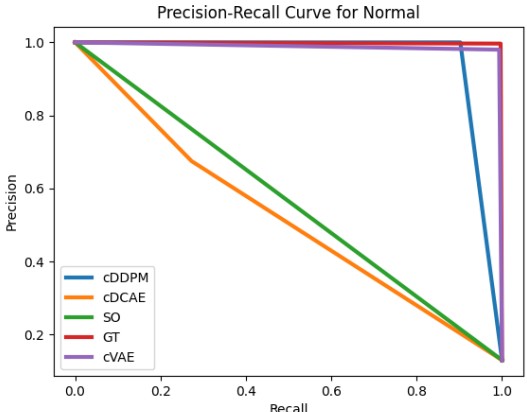

**Figure 2.** Precision Recall Plot for Normal class

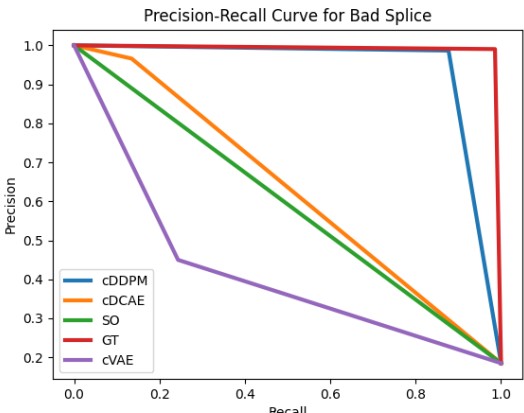

**Figure 3.** Precision Recall Plot for Bad Splice Class

cDDPM significantly outperform both SO and the cDCAE model, indicating that the model misclassifies instances in the test set as "Normal" far more often.

## 5.4 Bad Splice

**Table 4.** Performance metrics for Bad Splice

| Class | Precision | Recall | F1 Score |
|---|---|---|---|
| **Ground Truth** | 0.9905 | 0.9858 | 0.9882 |
| **Sub-Optimal** | 1.0000 | 0.0000 | 0.0000 |
| **cDDPM** | 0.9867 | 0.8771 | 0.9287 |
| **cDCAE** | 0.9661 | 0.1348 | 0.2365 |
| **cVAE** | 0.4498 | 0.2435 | 0.3160 |

It can be observed that again Ground Truth achieves the highest results, and the cDDPM generates the traces closes to the real traces most successfully. The cVAE here struggles to generate realistic traces for this class, achieving an F1 Score of only 0.3160. The cDCAE achieves poor results with an F1 Score of 0.2365. Both generative methods and the denoising method again prove more effective than training with noisy traces, however the cDDPM significantly outperforms the cVAE and cDCAE for this class with an F1 Score of 0.9287. The PR curve in 3 indicates that the cDDPM model performs very well, nearly matching the performance of Ground Truth, and significantly outperforms Sub-Optimal, cVAE and the cDCAE model. This visualization supports the use of cDDPM for generating high-SNR traces from low-SNR training data, demonstrating its effectiveness in preserving the quality of the generated data.

## 5.5 Generated Traces

A visualization of the Mean Absolute Distance from the Ground Truth can be observed Figure 11 in the Appendix. This is calculated by using the mean of Ground Truth traces of the "Normal" Class, where the SNR value is 30, and finding the mean difference between traces from each dataset of the same class and SNR value.

## 6 Conclusion

The cDDPM is capable of generating high quality denoised traces of fault classes despite not being trained on these samples. It records an F1 Score of higher than 0.9 for both classes, suggesting that the traces produced by the cDDPM are indistinguishable from the original traces contained in the test data. The conditional parameters enabled the model to infer what the samples would look like with a high SNR value. It is worth noting that due to the significant variability in the real world data used in this research both between and within the classes, observing the quality of the traces was difficult. This made it necessary measure the fidelity of the generated traces against the holdout dataset.

Traces generated by the cDDPM show a clear classification improvement over the noisy traces, the generative abilities of the cVAE and the traces denoised by the cDCAE on the same classes when all datasets were tested against the holdout test set. This proves the efficacy of the cDDPM to extrapolate samples which have not been seen by the model, or included in the training data. This work also highlights the efficacy of a cDDPM in generating 1-dimensional fault signals, which, as highlighted previously in Section 3, provides a significant contribution to the domain.

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

# 7 Appendix

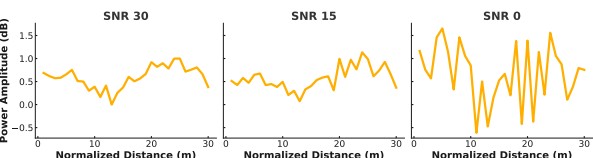

**Figure 6.** Signals from the Normal Class with different values of SNR. It can be seen that as value of the SNR increases the traces are smoother an of higher quality.

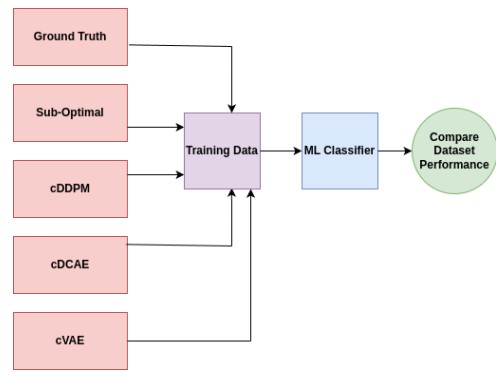

**Figure 4.** Workflow Diagram of Process

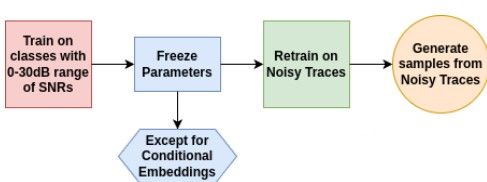

**Figure 5.** Training Process of cDDPM

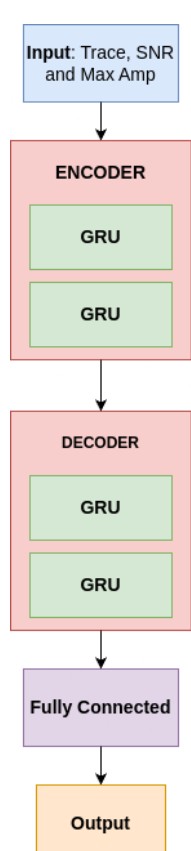

**Figure 7.** The ML Classifier Architecture.

**Table 5.** Number of samples in Ground Truth and Sub-Optimal Datasets

| Fault Type | # in GT | # in SO |
|---|---|---|
| **Normal** | 1142 | 2760 |
| **Bad Splice** | 1577 | 2545 |
| **Fiber Tapping** | 1606 | 1606 |
| **Dirty Connector** | 1622 | 1622 |
| **PC Connector** | 1587 | 1587 |
| **Reflector** | 1617 | 1617 |

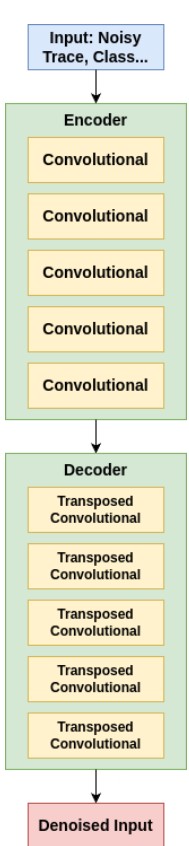

**Figure 8.** cDCAE Architecture

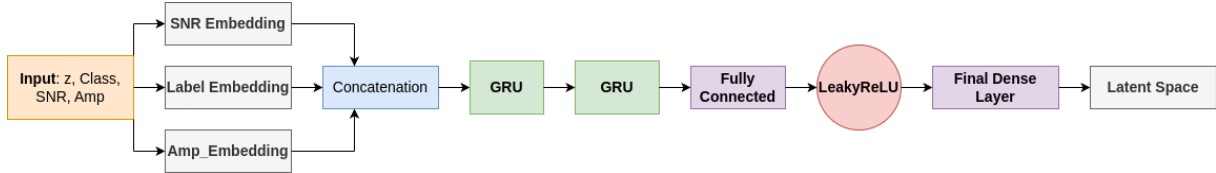

**Figure 9.** The cVAE Encoder.

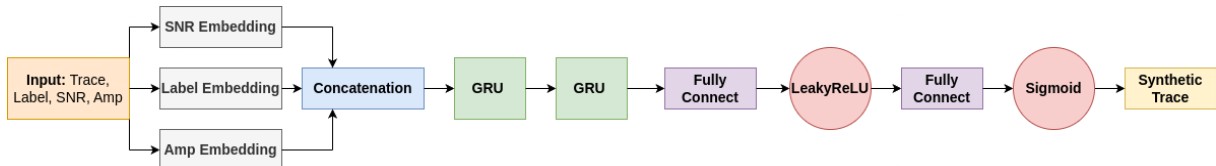

**Figure 10.** The cVAE Decoder.

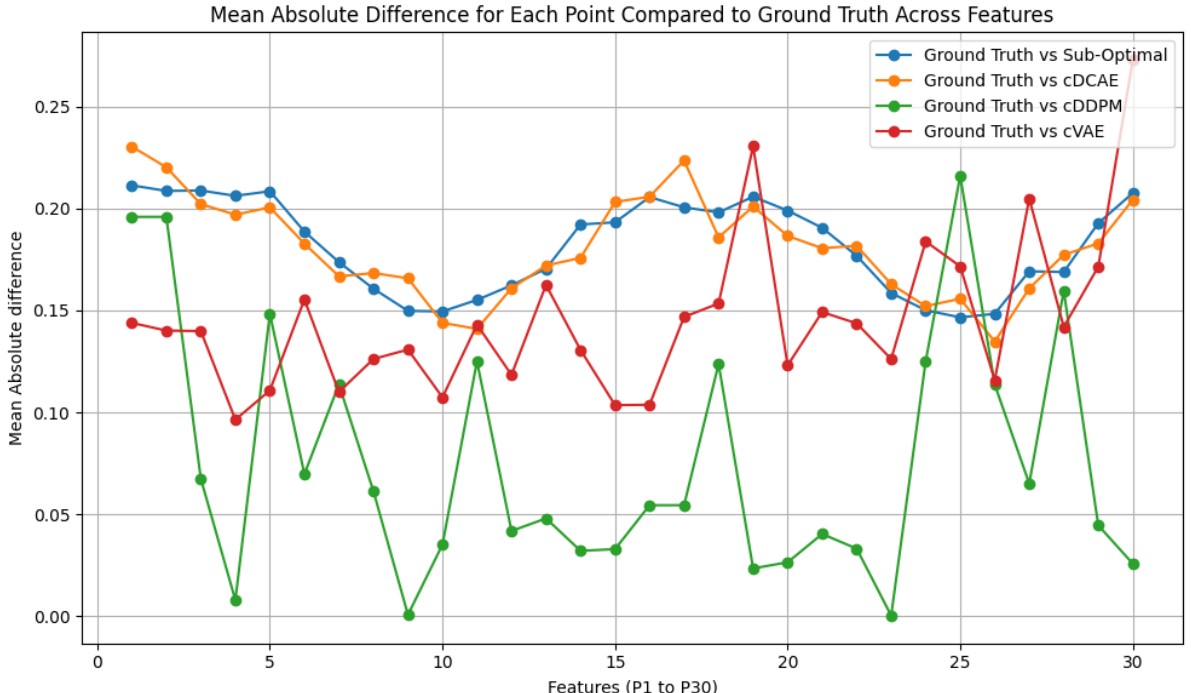

**Figure 11.** Mean Absolute Difference of All Datasets compared to Ground Truth

