# OpenReview forum: "Enhancing Fault Detection in Optical Networks with Conditional Denoising Diffusion Probabilistic Models"
_NLDL.org/2025/Conference — NLDL 2025 Poster_

### Official Review · Reviewer_oThH · 2024-09-20
**Review for "Enhancing Fault Detection in Optical Networks with Conditional Denoising Diffusion Probabilistic Models"**

**Confidence:** 3

**Summary:**

The paper proposes a data augmentation scheme for fault detection in optical networks via conditional denoising diffusion probabilistic models and compares the performance of a classifier trained on this data to one trained on the original data and data generated by a baseline autoencoder model.

**Strengths:**

The strengths of the paper are:
- Diffusion models for one-dimensional data are still fairly rare such that any progress in this direction is appreciated.
- The application case of Optical Networks is well-defined and compelling.
- The empirical results suggest that data augmentation with the proposed model may outperform several baselines.

**Weaknesses:**

However, I also see substantial room for improvement.
- Key aspects of the experiments are not clear; perhaps because terminology is inconsistently defined. In particular:
	- What are the 'Upper Bound' and 'Lower Bound' data sets? I assume these refer to Dataset A and B, respectively.
	- The results for which model are shown in Table 3?
	- Why is Table 4 labeled 'Per-class performance'? Isn't the per-class performance given in Table 3?
	- What are 'Original' and 'Noisy' in Table 4? Are these also the Upper and Lower Bound?
	- Why do 'normal' and 'bad splice' get separate sections but not the other classes? Are these classes particularly important? If so - why?
- In terms of interpretation: If Dataset A yields the best results on the validation data set - why generate any other data, at all? What would be the obstacle, in practice, to obtain this training data? How does the proposed model help to alleviate the problem?
- Relatedly: The text, at least at some places, seems to suggest that training data ideally contains only high SNR samples - but this does not seem representative of real-world data, such that such a classifier may overfit the training data. What am I overlooking, here?
- The denoising autoencoder is characterized as a discriminative model in line 304 of the paper. This strikes me as misleading. Autoencoders are not discriminative models. They might not be generate, either, but I would just cut the term 'discriminative' in this context.
- Relatedly: Why isn't a VAE used as a baseline? The architecture could remain the same as for the denoising autoencoder, but it would be substantially simpler to use the model as a generator.

**Final Rebuttal Confidence:**

3

**Final Rebuttal Justification:**

The revision clarified my questions and addressed my most important concerns. With these changes, I believe that the paper crossed the acceptance boundary.

**Justification:**

In its current stage, the experimental details strike me as too unclear to be convincing. Nonetheless, the paper may well have a solid core and I am happy to be convinced by a rebuttal.

---

> ### Author Rebuttal · Authors · 2024-10-25
>
> We thank the reviewers for their critiques and fair comments. We hope in the new iteration of our manuscript, we have complied with all of the suggestions.
> Main alterations from first iterations of the manuscript:
> - The abstract has been reformulated to more clearly explain the motivation of our approach
> - The introduction has been slightly altered to add more clarity
> - The Data Section has been altered to more clearly describe the counts and composition of each dataset and all inconsistent references have been removed
> - The Related Work section now includes a clear motivation to our work, explaing how our approach adds to the existing work on Diffusion Models
> - We have added a cVAE as a generative model baseline against the cDDPM on the recommendation of the reviewers. We also described in greater detail the role of the learnable conditional embeddings
> - We have added more clear description of the results and how they demonstrate the success of our methods
> - The following figures have been added in the Appendix:
> i) A Workflow Diagram of the process
> ii) A Process Diagram explaining how the cDDPM has been trained
> iii) Two diagrams describing the architecture of the cVAE
> iv) A Plot detailing the Mean Absolute Difference Between the Ground Truth dataset, and the other training datasets for a particular class and SNR value.
>
> 1) Thank you for you comment.
>       I) The Upper Bound data set refers to a data set where the training material have examples from the classes ‘Normal’ and ‘Bad Splice’ with high SNRs (above 20 dB). In the Lower Bound data set these classes only have samples with SNRs below dB. These have now been changed to make the paper more clear and consistent. The new names are Ground Truth dataset and Sub Optimal dataset
>
> ii) We admit that the original manuscript was unclear – also about the main goal of the experiments.  The major aim of the paper is to show that by using the conditional generative model, we can synthetically obtain examples with high SNRs for those two classes, despite we only would have low SNRs available. This is because it can “learn” the high SNR behaviour from the other classes. We then show that we can approach the performance of the Ground Truth data set when testing on our holdout dataset, which do have high SNR examples for all classes.  This is the main demonstration in the paper, and has now been clarified in the updated manuscript.  In other words, if we have some classes where the training examples have low SNRs we can boost this data set, so that it becomes significantly better to classify examples that actually have high SNRs.  This indicates that we take a low SNR trace and replace it with synthetic examples having high SNRs. So instead of denoising the specific trace, one can say that we replace the trace by synthetic examples of the same class having high SNR, which we then use for training.  The resulting performance on the hold out data set is an indication that we succeed in generating realistic examples with high SNRs.
>
> iii) Thank you for your comment. This has now been clarified in the manuscript to make  to a more intuitive name of Performance metrics.
>
> iv) Thank you for your comment. We agree that “Original”, “Noisy”, “Upper”, “Lower” and Dataset A/B is unclear. All inconsistent language has been removed and has been replaced with more intuitive wording – “Ground Truth” dataset and “Sub-Optimal” dataset where Ground Truth contains the “Ideal” traces of high SNR values between 20-30dB and Sub-Optimal dataset contains noisy traces for 2 classes (less than 5dB SNR).
>
> v) Thank you for your comment. The reason that these two classes are given subsections is that these classes, are the noisy traces in the Sub-Optimal dataset and used in different generative models as well as the denoising model in order to produce traces of better quality. We use these traces as they have the highest amount of noisy samples and therefore it is possible to perform our experiments on them.
>
> vi) We thank you for your comment. See also the answer to your first bullet point. The Upper Bound dataset (now called the Ground Truth dataset) represents the ideal case of having high SNRs available for ‘Normal’ and ‘Bad Splice” when training the model.  What we consider is a case where only noisy examples were available for these classes when training the model.  We then examine if we can boost the training data set by generating synthetic examples with high SNRs.
>
> vii) Thank you for your comment. You are right, and the discussions concerning this was not adequately listed in the manuscript. Our holdout data set contains real examples with high SNRs for all classes, because we wanted to test if we could transform low SNR training examples into synthetic ones with high SNRs that would resemble real ones to such a degree that the model could learn to classify the real ones correctly.  In a real scenario we should train both on the original examples with low SNRs as well as the generated ones with high SNRs.
> viii) Thank you for your comment. We agree and this has been removed from the manuscript.
>
> ix) Thank you for your comment. The cDCAEs used as a baseline because  of a preliminary work using this very same dataset and suggesting denoising the data using a cDCAE.  However, it is a good suggestion to extend it to using a VAE so that we also generate extra examples and we have done so, as another generative model baseline.

---

### Official Review · Reviewer_H4a2 · 2024-09-25
**Paper has some contribution but presentation and experimental evidence lacks clarity**

**Confidence:** 3

**Summary:**

The paper proposes the use of cDDPM (Denoising Diffusion Probablistic Model), i.e., a generative model, to augment a time series data set (i.e., optical time-domain reclectogrammy traces). The model learns signal characteristics from high SNR anomaly data samples, which are less affected by noise. On the resulting data sets a classifier can achieve higher accuracy, precision and recall.

**Strengths:**

Strengths:
- The paper uses publicly available data sets

**Weaknesses:**

Weaknesses:
- While writing and gammar is generally good, the paper is unclear at many points and I had to read it several times to uncover the actual contribution and how everything fits together
- There is a lot of unnecessary information and figures that can be omitted in order to gain space for relevant descriptions
- The paper discusses a lot of research that seems to be directly competitive to the proposed methods but only one method serves as a baseline (and it is unclear if it is one of the related works)
- The experimental results (especially those in 5.1 and 5.2 are not discussed.

**Justification:**

Questions/Remarks:
- I found it very hard to understand the actual contribution of the paper after reading the introduction. My suggestion is to add a figure in the intro to sketch what the paper is about
- Section 2: seems to be unnecessary to me. What I would have lover more would be a formal introduction of the data samples and the prediction tasks, i.e., how the data samples are arranged and used as tensor for computation (and later when explain the loss function this would be beneficial)
- Figure 1. Should be a vector graphic
- There is a lot of related work in section 3. Why is baseline comparison in the experimental section so limited?
- Line 217: "This dataset is comprised of both the real traces from the aforementioned four classes " - which four classes are you referring to? Please make it clear how the data sets are built, the definitions are currently a bit sloppy
- Line 280: "Using embeddings creates a learnable representation for each condition" - what does this mean?
- Section 4.3.4: I would have liked to see a mathematical definition of the loss function. It is hard to re-implement everything based on the current information
- Table 2: show the results for the datasets separately - are there any noticeable things?
- Table 2/Section 5.2: what is upper bound and lower bound? How are they computed? In Section 5.3 you talk about the upper bound data set. This however has at no point been defined.
- What's the benefit of Figs. 3 and 4?
-

Some minor points:
- line 052 ", i.e."
- line 075 training appears twice
- line 130 increase[-s-]
- line 149 that appears twice
- line 159 reference missing
- line 177 model[s]
- line 227: we design a[n] ML
- line 231 is comprised an - word missing
- line 233: layer[s]
- line 301: Abdelli et.al

---

> ### Author Rebuttal · Authors · 2024-10-25
>
> We thank the reviewers for their critiques and fair comments. We hope in the new iteration of our manuscript, we have complied with all of the suggestions.
> Main alterations from first iterations of the manuscript:
> - The abstract has been reformulated to more clearly explain the motivation of our approach
> - The introduction has been slightly altered to add more clarity
> - The Data Section has been altered to more clearly describe the counts and composition of each dataset and all inconsistent references have been removed
> - The Related Work section now includes a clear motivation to our work, explaing how our approach adds to the existing work on Diffusion Models
> - We have added a cVAE as a generative model baseline against the cDDPM on the recommendation of the reviewers. We also described in greater detail the role of the learnable conditional embeddings
> - We have added more clear description of the results and how they demonstrate the success of our methods
> - The following figures have been added in the Appendix:
> i) A Workflow Diagram of the process
> ii) A Process Diagram explaining how the cDDPM has been trained
> iii) Two diagrams describing the architecture of the cVAE
> iv) A Plot detailing the Mean Absolute Difference Between the Ground Truth dataset, and the other training datasets for a particular class and SNR value.
>
> 1) Thank you for your comment. We agree that in the original iteration many of the terminology and description of the experimental process was unclear. We have removed unclear terminology such as Upper Bound/ Lower Bound datasets and Dataset A/B in order to emphasise our approach. We have also reformulated the abstract  to clarify the scope.
>
> 2) Thank you for your comment. We have updated and shortened each of the opening section for clarity and brevity. We have updated the language to show more consistency and we have clarified the unclear descriptions in the results section in order to stress the benefits of our approach.
>
> 3) Thank you for your comment. We have added another generative model the cVAE and stressed the necessary baselines in the related work section. We have also emphasized why our approach has a clear motivation from previous OTDR related anomaly detection in the related work section and how diffusion models can help bridge this gap in the literature.
>
> 4) Thank you for your comment. We agree that in the previous iteration of our paper the results were not discussed to a sufficient degree. We have added relevant text in the results section that we believe lends more clarity to the benefits of our approach.
>
> Questions/ Remarks
>
> 1) Thank you for your comment. We agree that in the original contribution there was inconsistent language which lead to confusion for the reader and have made the necessary changes. We have added a relevant sketch of the workflow of our approach in order to better motivate our proposed approach. We have also edited each section for more clarity.
>
> 2) Thank you for your comment. We agree that the original contribution lacked some clarity on these points. We have added a more direct analysis, of how exactly the data was arranged in Section 2 and added the correct mathematical equation for the loss function in section 4.3.4
>
> 3) Thank you for your comment. This has been changed in the manuscript.
>
> 4) Thank you for your comment. We have added a cVAE as another baseline in order to further emphasize the success of our proposed model. We have made the related work section more succinct and have added clarity that extends the previous work on both anomaly detection using OTDR data and previous work on Diffusion models.
>
> 5) Thank you for your comment. We agree and have added further clarity in section 4.1 to make it more readable. We have also endeavoured to make the language consistent so that our approach become more clear.
>
> 6) Thank you for your comment. We have clarified in the manuscript what is meant by this and how it benefits our approach in section 4.3.3. We explain the learnable embeddings are “ initialized as random noise and optimized during training via backpropagation. This allows the model to learn relationships between different conditions.” We stress this in the manuscript as it benefits our approach of first training the data on the entire range of SNR traces and then inferring how traces with high SNR values would look, from using only noisy samples. We motivate our approach by the methods in paper [1] and cite this in the manuscript.
> [1] De Donno, C., Hediyeh-Zadeh, S., Moinfar, A.A. et al. Population-level integration of single-cell datasets enables multi-scale analysis across samples. Nat Methods 20, 1683–1692 (2023). https://doi.org/10.1038/s41592-023-02035-2
>
> 7) Thank you for your comment. We agree and have added this in section 4.3.4.
>
> 8) Thank you for your comment. We have endeavoured to make this section more readable, and clarify exactly why the counts of these classes are as they are. For “PC Connector”, “Dirty Connector”, “Fiber Tapping” and “Reflector” the counts of each class does not change between datasets, the same samples are used. For “Bad Splice” and “Normal” these classes are used in the generative process to generate better quality samples.
>
> 9) Thank you for your comment. We agree that in the original contribution, there was a lot of inconsistent language. All references to Upper and Lower Bound and datasets A and B have now been removed from the manuscript and replaced by two more intuive names “Ground Truth” and “ Sub-Optimal”. “Ground Truth” contains traces with only high SNR values for all 6 classes and serves to illustrate that the Model performs well under conditions of high quality samples. The sub-optimal dataset contains of high SNR (20-30dB) values for classes  “PC Connector”, “Dirty Connector”, “Fiber Tapping” and “Reflector” and low quality SNR traces (<5dB) for “Normal” and “Bad Splice” classes. This dataset is used to stress the importance of using data of sufficiently high quality in order to make accurate fault detection. We agree that this was inadequately explained in the original iteration.
>
> 10) Thank you for your comment. We agree that this was not sufficiently explained in the original iteration. Figures 3 and 4 are “one-vs-rest” Precision-Recall curves for the two classes being analyzed. The curves have each fo the training datasets – Ground Truth, Sub-Optimal, cVAE, cDDPM and cDCAE compared with each other to assess which best can classify the holdout test set. We have endeavoured to explain this more clearly in the manuscript.
>
> 11) Thank you for your attention to detail. We have amended these errors in the manuscript.

---

### Official Review · Reviewer_KPoU · 2024-09-26
**Interesting application of DDPMs for fault detection in optical fibers**

**Confidence:** 4

**Summary:**

This work is focused on the case of fault detection in optical fibers based on reflectometry data. Noise can make it challenging to detect certain anomalies in this data, and the paper proposes to unse generative models to enhance the signal-to-noise (SNR) ratio. Specifically, a conditional denoising diffusion probabilistic model is used to generate synthetic data with improved SNRs. The proposed solution is evaluated on relevant datasets and compared with suitable baselines, with promising results.

**Strengths:**

1) The application of denoising diffusion probabilistic models to fault detection in optical networks seems new, and it is interesting to see such models applied with success also in this domain.

2) The methodology is evaluated on relevant datasets with promising results.

3) The writing is mostly clear and easy to follow.

**Weaknesses:**

1) The baselines that are reported in the manuscript are suitable. However, it would be beneficial to also include baselines that are not based on neural networks. Many such baselines are available (see e.g. [1]), and would give the reader a better impression of the potential of proposed approach and need for using complex deep learning-based methods.

2) The referencing of the paper could be improved. Many works are mentioned but not properly cited. For example, variational auto encoder, Wasserstein GAN, GRU, LeakyReLU, and Dropout are not properly referenced. Even thought these are widely used methods and components, it is till important to give reference them properly. Furthermore, the references themselves can also be improved. Several citations now reference the pre-print version when published versions exist. This should be updated.





[1] Anomaly detection in time series: a comprehensive evaluation. Proceedings of the VLDB Endowment, Volume 15, Issue 9
Pages 1779 - 1797 https://doi.org/10.14778/3538598.3538602. Sebastian Schmidl, Phillip Wenig, Thorsten Papenbrock.

**Final Rebuttal Confidence:**

4

**Final Rebuttal Justification:**

I think the authors have done a good job in revising the manuscript, and I think the new version is an improvement. Given that my impression was positive even before the revision, I will keep my recommendation from the previous phase.

**Justification:**

The technical novelty of this paper is not particularly significant. But the application is novel and interesting, and I think it fits the conference well. The paper is mostly well-written and the results are promising. There is potential for improvements both in terms of adding one or more non deep learning-based baselines and improving the use of references, but I do not see these as critical flaws. Therefore, I recommend to accept the paper.

---

> ### Author Rebuttal · Authors · 2024-10-25
>
> We thank the reviewers for their critiques and fair comments. We hope in the new iteration of our manuscript, we have complied with all of the suggestions.
> Main alterations from first iterations of the manuscript:
> - The abstract has been reformulated to more clearly explain the motivation of our approach
> - The introduction has been slightly altered to add more clarity
> - The Data Section has been altered to more clearly describe the counts and composition of each dataset and all inconsistent references have been removed
> - The Related Work section now includes a clear motivation to our work, explaing how our approach adds to the existing work on Diffusion Models
> - We have added a cVAE as a generative model baseline against the cDDPM on the recommendation of the reviewers. We also described in greater detail the role of the learnable conditional embeddings
> - We have added more clear description of the results and how they demonstrate the success of our methods
> - The following figures have been added in the Appendix:
> i) A Workflow Diagram of the process
> ii) A Process Diagram explaining how the cDDPM has been trained
> iii) Two diagrams describing the architecture of the cVAE
> iv) A Plot detailing the Mean Absolute Difference Between the Ground Truth dataset, and the other training datasets for a particular class and SNR value.
>
> 1) Thank you for your comment. We agree that there was some confusion regarding baselines in the original manuscript. We use the ML classifier here as a similarity metric to assess how high quality or low quality traces  affect the performance of this classifier. We agree that the inclusion of more baselines for generating denoised samples would be beneficial. We have added a conditional Variational Autoencoder as a baseline to increase the quality of the SNR values of our traces. We also use an additional visualization in the appendix where we describe the Mean Absolute Difference between all our training datasets and the Ground Truth datasets.
>
> 2) Thank you for your comment. We agree and have updated the referencing accordingly.

---

### Official Review · Reviewer_jhSS · 2024-10-10
**While the authors conduct research on an interesting and relevant topic, the research lacks clarity, making it difficult to understand the relevance of the results.**

**Confidence:** 4

**Summary:**

The authors consider the case where data scarcity poses a challenge to establish good defect detection in optical fibers. They propose generating synthetic data with improved SNRs to train better performing anomaly classifiers.

**Strengths:**

The authors consider a relevant topic from the machine learning and use case point of view.

**Weaknesses:**

While the manuscript touches on a relevant topic, we consider it should be better structured and, in particular, clearer regarding the experimental side and results obtained. Below, we provide details regarding some improvement opportunities.

GENERAL COMMENTS

   (1)- We would appreciate it if the authors could provide a brief description regarding the meaning of each class and the implications for the optical networks maintenance. Are all errors equally critical? Is it relevant only to identify the defect that took place or that such a defect will likely take place in the future? Furthermore, when presenting the results: (i) are results differences between models statistically significantly different? (ii) what are the practical implications of the results (e.g., detecting some defects very well, not that well some others).

   (2)- The authors aim to enhance the classifiers' performance by generating synthetic data. (i) how much data do they generate (from each class)? (ii) how does the amount of synthetic data impact the discriminative performance of the model? (iii) how similar is the synthetic data to the original one? (iv) what metric should be used to assess the similarity? (v) could the authors provide visual examples of real and synthetic data?, (vi) do the authors venture some explanation as to why less noisy data results in a better classifier?

   (3)- Why did the authors choose a holdout dataset and not cross-validation? How was the holdout dataset created? May the authors provide some insights on whether the data distribution between train and test sets is similar or different?

   (4)- Metrics: (i) we encourage the authors to provide AUC ROC scores for their models to have a threshold-independent estimate of the models' quality. (ii) how did the authors determine the cut-off thresholds when evaluating accuracy/precision/recall/F1?

   (5)- The research paper aims to enhance the classifiers' performance by generating synthetic data. We encourage the authors to more clearly specify the experiments performed, how these relate to the overall goal, and what the particular hypothesis/goals are motivating that experiment. The authors should report the results in the same structure, showing correspondence between hypotheses and experiments and the experiment outcomes. Furthermore, we invite them to decouple the general methodology section from the experiments performed and the descriptions of the particular models, and provide some figures describing in detail the whole methodology/procedure. Do we use a single classifier? Do we use many of them? Why?

   (6)- There is a lack of clarity regarding the datasets used to perform the experiments. In particular, the authors mention upper and lower-bound datasets in the Introduction and later introduce datasets 1-4 in subsection 4.1. A figure or table would be helpful to explain more clearly the characteristics and interplay between datasets 1-4 and upper and lower bound datasets: which classes (e.g., "Bad splice") do they have? how many instances? is there a dB segmentation? which ones include synthetic data? what is the role of the cDDPM and cDCAE models?

   (7)—Section 5.1.1: It is not clear to us what the authors meant regarding the results reported in Table 2. While the caption mentions that accuracy corresponds to performance regarding three training datasets, they report on upper/lower bounds and two models.

   (8)- The authors devote subsection 5.4 to "Bad splice" - why does this kind of fault merit a whole subsection, while the rest seems to not be reported in detail?

   (9) Figure 2 introduces three kinds of embeddings: class embeddings, SNR embeddings, and Amp embeddings. We encourage the authors to provide further details on the rationale behind selecting these conditions and how these embeddings are computed and how they relate to the cDDPM and cDCAE models.


  FIGURES

   (10)- Figure 2: it seems some arrows are missing from two embedding boxes

  TABLES

   (11)- Align numbers to the right and use the same number of decimals. We suggest using four decimals, given many cases reach the same result when considering a two-decimal precision.

   (12)- Table 4: while the paper is about defect detection, why do the authors report (only) about the Normal class? There seems to be a mismatch between the table and the caption.

  MINOR COMMENTS

   (13)- "with four classes having training training" -> "with four classes having training training"

   (14)- "linear layer has a leakyRelu()" -> "linear layer has a leaky ReLU"

   (15)- "1.1 Background" -> the subsection is redundant - no other subsections for Introduction exist.

**Final Rebuttal Confidence:**

4

**Final Rebuttal Justification:**

The authors have addressed all of the items and improved the manuscript.

**Justification:**

While the approach and use case are relevant, the paper lacks clarity regarding the experimental setup and the results, making it difficult to assess the contribution of using a particular approach to generate synthetic data to enhance the classifier's performance.

---

> ### Author Rebuttal · Authors · 2024-10-25
>
> We thank the reviewers for their critiques and fair comments. We hope in the new iteration of our manuscript, we have complied with all of the suggestions.
> Main alterations from first iterations of the manuscript:
> - The abstract has been reformulated to more clearly explain the motivation of our approach
> - The introduction has been slightly altered to add more clarity
> - The Data Section has been altered to more clearly describe the counts and composition of each dataset and all inconsistent references have been removed
> - The Related Work section now includes a clear motivation to our work, explaing how our approach adds to the existing work on Diffusion Models
> - We have added a cVAE as a generative model baseline against the cDDPM on the recommendation of the reviewers. We also described in greater detail the role of the learnable conditional embeddings
> - We have added more clear description of the results and how they demonstrate the success of our methods
> - The following figures have been added in the Appendix:
> i) A Workflow Diagram of the process
> ii) A Process Diagram explaining how the cDDPM has been trained
> iii) Two diagrams describing the architecture of the cVAE
> iv) A Plot detailing the Mean Absolute Difference Between the Ground Truth dataset, and the other training datasets for a particular class and SNR value.
>
> 1) Thank you for your comment. We acknowledge that the questions raised here are relevant in a practical setting with respect to fault classification and predictive/preventive maintenance issues. However, the aim of our research has the more general scope of showing, with real OTDR traces, that the SNR level of the traces can be improved using a conditional Denoising Diffusion Probabilistic Model. We also add that we have reformulated the abstract to better motivate and clarify the scope of our approach.
>
> 2) Thank you for your comment.
>  i)For the two generative methods (cDDPM and cVAE), we generate 1600 samples for the  'Normal' and 'Bad Splice' classes  as this is the mean of the remaining classes, where we  use the original traces having high SNRs. For the cDCAE, the denoised versions of all the traces for “Normal” and “Bad Splice” under 5dB are used. These are 2760 an 2545 respectively. This  has now been clarified in the manuscript.
> ii) We use the sub-optimal dataset to demonstrate the importance of high quality traces. Then we show through use of the cDDPM, and other baselines how the denoised traces of higher quality.
> iii) The way we “prove” and “test” the similarity of the generated  high SNR examples to real ones is that we use each generated dataset as training data to be tested against a holdout dataset in the ML classifier. We prove that the cDDPM leads to favourably comparative results with the original data, referred to as “Ground Truth”. This has now been reworded in the manuscript for greater clarity.
> iv) The ML classifier is used as a similarity metric in this case because in order to obtain good performance on the holdout data set for the 'Normal' and 'Bad Splice'  classes the generated  samples must mimic the real traces with high SNRs. We observe that by using the synthetic generated examples we come close to the performance obtained using Ground Truth samples from these classes. The classifier is tuned to the same hyperparameters and each dataset is used separately as training data and the performance is compared against the same holdout set.
> v)  We note that there is a significant amount of variability between the traces within each class in a real world dataset, hence motivating the need for automating this process. However we have added a figure in the Appendix that shows the Mean absolute difference from each dataset to the Ground Truth dataset for the Normal class, which we believe illustratively describes our results
> vi) Less noisy data, or data with a higher SNR value, results in a better classifier as often anomalies in the traces can be indistinguishable from the noise present in low quality samples. We have clarified this in the manuscript.
>
> 3) Thank you for your comment.
> I) Actually the holdout dataset works as a test set. The main idea is that the test set contains examples from the classes ‘Normal’ and  ‘Bad Splice’ that have SNRs higher than 20dB, whereas the Sub-Optimal set before augmenting the synthetic examples only have examples in those classes with SNRs less than 5dB.  The idea being that by replacing the Sub-optimal set with synthetic examples having high SNRs one would hope to get a better classification model for the real examples having those high SNRs.
> ii) The holdout dataset was created by sampling 450 samples from each of the 6 classes at the start of the process, with SNR levels between 20-30dB.
> Iii) The data distribution between the training and test sets is not significantly different. The test set is created with 450 samples from each class. Each training set from the generative models is created with approximately 1600 samples per class in an effort to provide a balanced training set. The exception of this is the Sub-Optimal or “Noisy” training set, in which the samples less than 5dB are included as training data, or the cDCAE training data, in which the denoised samples, less than 5dB are included as training data, each with over 2500 samples.
>
> 4) Thank you for your comment.
> i) We actually show Precision-Recall Threshold curves for one class against the rest for the “Normal” and “Bad Splice” classes. This is because they are the only classes generated or otherwise denoised. The other classes are used in the generative models in order to “learn” what traces “look like” with high SNR values.
> ii) The accuracy, precision, recall and F1 scores are calculated as a result of multiclass classification, and every sample is classified according to the class output note with the highest output.
>
> 5) Thank you for your comment.
> I) We acknowledge that in the first iteration of our contribution, the scope of our paper was unclear and major alterations have now been made to deal with that.
> ii) We prove the importance of SNR values to the overall improvement as we test the classifier using a sub-optimal set of only noisy samples in these two classes as part of our experiments. Although these samples have the same fault or class label as the samples in the holdout test set, they fail to achieve high performance when being tested in the classifier, showing the importance of using samples with high SNR values.
> iii) We use a single classifier tuned with the same hyperparameters and train it with different training datasets, each with the same class labels, and discover that the cDDPM synthesized data produces the best result, with the exception of the Ground Truth training set.
> iv) We added a figure describing the overall concept to the manuscript and endeavoured to make our overall aim and experiments more clear.
>
> 6) Thank you for your comment
> I) We have removed all references to upper and lower bound from the manuscript and instead replaced them with more intuitive names – Ground Truth and Sub-Optimal. We have described these in the data section in order to add more clarity
> ii) We have added clarity on the dB segmentation in the manuscript
> iii) The role of the cDDPM and cDCAE models is to generate denoised data or denoise the existing data respectively
>
> 7) Thank you for your comment. This has now been clarified in the manuscript. The role of this table is to compare the overall accuracy between the different training sets used in the same ML classifier and tested against the same holdout dataset. Again, we have removed all inconsistent language referring to upper and lower bounds for the sake of clarity
>
> 8) Thank you for your comment. The reason is that in our scenario the ‘Normal’ and ‘Bad splice’ classes  are made to represent  the two classes, where the training material only contain noisy  examples.  Therefore their classification accuracies are low. We then replace with synthetic examples to counteract this deficiency, and specifically study the effect that this has on the classifiers performance. This is compared to the case where we actually have real training examples with high SNRs (previously called the upper bound model and now called Ground Truth).
>
> 9) Thank you for your comment. This has now been clarified in the manuscript. The class, SNR and Amp embedding are chosen due to the variability in each class, and within each class in the manifestation of traces. Each embedding shows how the traces can appear differently in each class and all interact with each other to produce traces from each conditional embedding.
>
> 10) This has now been fixed in the manuscript. With thanks.
>
> 11) This has now been fixed in the manuscript. With thanks.
> 12) We acknowledge it can seem confusing that  we have focus on the "Normal  Class" (as well as the "Bad Splice") since the scope of the work is anomaly detection.  However, in order to test our methodology of generating high SNR traces for classes with only low SNR traces we needed classes that had examples over the whole range of SNRs. Thereby we can also have a hold out data set with REAL traces with high SNR to  test against. It appeared that the "Normal" class together with the "Bad splice" classes fulfilled these requirements.
>
> 13) All minor comments have been adjusted in the manuscript. Thank you for observing and for your close examination.

---

### Meta-Review · Area_Chair_gHj2 · 2024-10-30

**Recommendation:** Accept (Poster)
**Confidence:** 4

**Metareview:**

The paper explores the area of fault detection in optical networks using reflectometry data. They utilised conditional Denoising Diffusion Probabilistic Models (cDDPM) to generate synthetic data with high signal-to-noise ratio (SNR) for classes with low-quality samples.
They compared the augmented dataset with a baseline dataset by training an anomaly classifier and evaluating its performance on a hold-out dataset with high-quality traces.

The domain is rather interesting and the results are promising. The technical novelty is rather limited though.
Nevertheless, the reviewers were positive, therefore the paper could be accepted and discussed at the conference.

**Suggested Changes To The Recommendation:**

1: I agree that the recommendation could be moved down

---

### Decision · Program_Chairs · 2024-11-06

**Decision:**

Accept (Poster)

**Comment:**

We recommend a poster presentation given the AC and reviewers recommendations.